# Quantum-Based Modeling of Dephosphorylation in the Catalytic Site of Serine/Threonine Protein Phosphatase-5 (PPP5C)

**E. Alan Salter [1,2], Andrzej Wierzbicki [1] and Richard E. Honkanen [2,*]**

[1] Department of Chemistry, University of South Alabama, Mobile, AL 36688, USA; asalter@southalabama.edu (E.A.S.); awierzbicki@southalabama.edu (A.W.)

[2] Department of Biochemistry and Molecular Biology, College of Medicine, University of South Alabama, Mobile, AL 36688, USA

[*] Correspondence: rhonkanen@southalabama.edu; Tel.: +1-251-460-6402

**Abstract:** Serine/threonine protein phosphatase-5 (PP5; PPP5C) is a member of the phosphoprotein phosphatase (PPP) gene family. The PPP catalytic domains feature a bimetal system ($M_1/M_2$), an associated bridge hydroxide ($W^1(OH^-)$), an $M_1$-bound water/hydroxide ($W^2$), and a highly conserved core sequence. The PPPs are presumed to share a common mechanism: The seryl/threonyl phosphoryl group of the phosphoprotein coordinates the metal ions, $W^1(OH^-)$ attacks the central phosphorous atom, rupturing the antipodal phosphoester bond and releasing the phosphate-free protein. Also, a histidine/aspartate tandem is responsible for protonating the exiting seryl/threonyl alkoxide. Here, we employed quantum-based computations on a large section of the PP5 catalytic site. A 33-residue, ONIOM(UB3LYP/6-31G(d):UPM7) model was built to perform computations using methylphosphate dianion as a stand-in substrate for phosphoserine/phosphothreonine. We present a concerted transition state (TS) in which $W^1(OH^-)$ attacks the phosphate center at the same time that the exiting seryl/threonyl alkoxide is protonated directly by the $His^{304}/Asp^{274}$ tandem, with $W^2$ assigned as a water molecule: $W^2(H_2O)$. $Arg^{275}$, proximal to $M_1$, stabilizes the substrate and TS by binding both the ester oxygen ($O^\gamma$) and a phosphoryl oxygen ($O^1$) in a bidentate fashion; in the product state, $Tyr^{451}$ aids in decoupling $Arg^{275}$ from $O^1$ of the product phosphate ion. The reaction is exothermic ($\Delta H = -2.0$ kcal/mol), occurs in a single step, and has a low activation barrier ($\Delta H^\ddagger = +10.0$ kcal/mol). Our work is an improvement over an earlier computational study that also found bond rupture and alkoxide protonation to be concerted, but concluded that $Arg^{275}$ is deprotonated during the reactant and TS stages of the pathway. In that earlier study, the critical electron-withdrawal role that $Arg^{275}$ plays during the hydroxide attack was not correctly accounted for.

**Keywords:** ONIOM calculations; DFT; protein dephosphorylation; enzyme catalysis; metalloenzymes; phosphoester hydrolysis; transition state

## 1. Introduction

Protein phosphorylation increases the functional diversity of the proteome and is critical to the regulation of numerous cellular processes. About 13,000 human proteins have one or more "p-sites"—amino acid residues where reversible phosphorylation occurs [1]. The presence or absence of a phosphoryl group at a p-site impacts protein structure, enzyme activity, interactions with other proteins, transport properties, and, ultimately, cell function and metabolism [2]. Countering the action of phosphoryl attachment by protein kinase enzymes (phosphorylation), the protein phosphatases catalyze the hydrolysis that liberates a phosphate ion and restores the p-site residue (dephosphorylation). The hydroxyl groups of serine, threonine, and tyrosine are amenable to phosphoryl attachment via

a phosphoester bond, and most p-sites are one of these three amino acids, with serine dominant, threonine second, and tyrosine a distant third [3].

Our interest is the common mechanism by which the members of the phosphoprotein phosphatase (PPP) gene family (PPP1C, PPP2C, PPP3C/calcineurin, PPP4C, PPP5C, PPP6C, and PPPEF/PP7), a subcategory of the broader class of protein serine/threonine phosphatases, catalyze dephosphorylation at seryl and threonyl p-sites. The PPPs share a highly conserved catalytic core [4] featuring a bimetal system ($M_1$/$M_2$) as the site of substrate binding and hydrolysis. In this study, we focused on serine/threonine protein phosphatase-5 (PP5) as representative of the PPP family. PP5 expression is known to affect cell proliferation [5,6] and stress-induced responses [7–9], and its overexpression is implicated in human breast carcinoma [10], non-small cell lung cancer [11], and other cancers [12].

Co-crystal structures of PP5/phosphate [13], PP1/tungstate [14], and PP1/phosphate [15], reveal common PPP catalytic-site features, including: A bimetal system held in place by the same amino-acid scaffold of six residues, a bridge hydroxide/water ($W^1$), a metal-bound phosphate (or tungstate) positioned by the same four residues, and a cooperative histidine/aspartate tandem. As shown for PP5/phosphate, Figure 1, the alignment of the phosphate with $W^1$ strongly implies a nucleophilic attack on the P center by $W^1$, with subsequent protonation of the exiting alkoxide at the $O^4$ site by $H^+$ transfer from $N^\epsilon$ of His$^{304}$. Furthermore, Swingle et al. argued that $W^1$ is a hydroxide ion, and that the exiting alkoxide oxygen is probably not metal bound [13]. Of course, this inferred mechanistic scenario depends on a match between the phosphate ion's binding mode seen in the PP5 co-crystal and the binding mode of the phosphoserine/phosphothreonine substrate, which is unknown. Notably, the co-crystals also reveal a second water/hydroxide ($W^2$) coordinated to $M_1$, a backbone carbonyl (PP5:His$^{427}$) positioned to direct $W^1$, and a backbone -NH group (PP5:Phe$^{315}$) anchoring the His/Asp tandem at the backend, Figure 1. In PP5, Asp$^{274}$ is also anchored by the nonconserved Asn$^{310}$ (likewise in PP1 and PP7, threonine otherwise). Thus, these co-crystals, along with mutation studies [16,17] and analysis, established a plausible pathway for PPP dephosphorylation [13–15]. The substrate's anticipated mode of binding to $M_1$/$M_2$ and the $W^1$(OH$^-$) attack on the P center were later corroborated by density functional theory (DFT) calculations by Ribeiro et al. on a cluster model of PP5/phosphoserine [18].

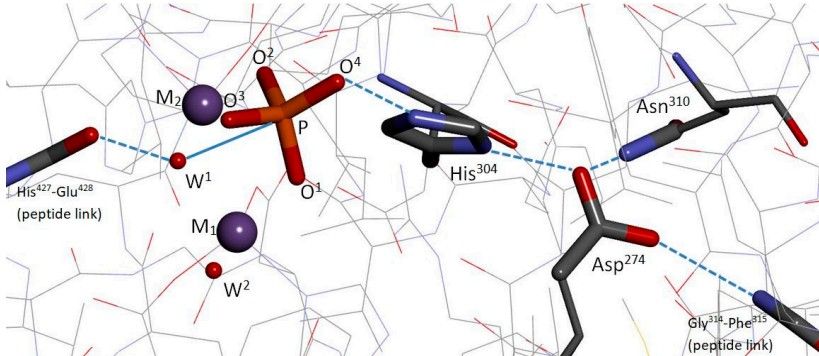

**Figure 1.** The PP5/phosphate co-crystal (pdb entry 1S95 [13]) and its implications for phosphoester hydrolysis. The phosphate ion's $O^1$ and $O^2$ are coordinated to two Mn$^{2+}$ ions, $M_1$ and $M_2$, respectively. $W^1$ is the oxygen of what is now accepted to be a bridge hydroxide. The hydroxide is oriented by the peptide carbonyl at left and is positioned along the P-$O^4$ axis (solid line) for a backside attack, theoretically, on the central P atom of a similarly bound phosphoester. $N^\epsilon$ of His$^{304}$ is almost certainly protonated, and $O^4$ marks what would be a seryl/threonyl $O^\gamma$ requiring a proton as the phosphoester bond breaks. The His$^{304}$/Asp$^{274}$ tandem is coupled by an H-bond, and Asp$^{274}$ is additionally H-bonded to Asn$^{310}$ and the -NH moiety of the peptide link at right. $W^2$ is the oxygen of a water/hydroxide ligated to $M_1$. Dashed lines indicate H-bonds.

In their PP5 study [18], Ribeiro et al. presented pathways for two different systems, the first of which they endorsed: (I) $W^1$(OH$^-$)/$W^2$(OH$^-$) system (one step, exothermic): Arg$^{275}$ loses a proton

to the $W^2(OH^-)$ pre-reaction. That is, $W^2$ is actually a water and $Arg^{275}$ is deprotonated during the reactant and TS stages of the pathway; the proton is returned to $Arg^{275}$ in the product state. The seryl alkoxide is protonated by direct $H^+$ transfer from $His^{304}$ as $W^1$ attacks. (II) $W^1(OH^-)/W^2(H_2O)$ system (two steps, endothermic): The seryl alkoxide is protonated by $Arg^{275}$ in step 1, in which the $W^1(OH^-)$ attack also occurs. In step 2, $H^+$ is transferred from $His^{304}$ to $Arg^{275}$ through the serine's alcohol group.

Both pathways (I) and (II) contain modeling artifacts. The Ribeiro et al. model system comprised 10 amino acid moieties + $M_1/M_2$ + $W^1(OH^-)$+ $W^2$ + phosphoserine. The amino acid moieties were terminated by frozen methyl groups. $Asn^{310}$ and the $Gly^{314}$-$Phe^{315}$ peptide linkage, which together bind $Asp^{274}$ (see Figure 1), $Asp^{388}$, which binds $Arg^{400}$ (see Figure 2), and $Tyr^{451}$, which abuts and interacts with $Arg^{275}$, were not represented. Consequently, the optimized positions of $Asp^{274}$, $Arg^{400}$, and $Arg^{275}$ diverged significantly from their native PP5/phosphate co-crystal positions. It should be added that a molecular dynamics (MD) study of PP5/phosphoserine by Wang and Yan [19] concluded that $W^1(OH^-)$-based systems are stable, while $W^1(H_2O)$-based systems are not, in further support of the conclusion that $W^1$ is a hydroxide. Additionally, they found that the $W^1(OH^-)/W^2(H_2O)$ system has greater substrate affinity versus the $W^1(OH^-)/W^2(OH^-)$ system. However, $Arg^{275}$ and $W^2$ were apparently not modeled in the latter system, as would be prescribed for the reactant stage of pathway (I), with $Arg^{275}$ as a neutral species and $W^2$ as a water molecule.

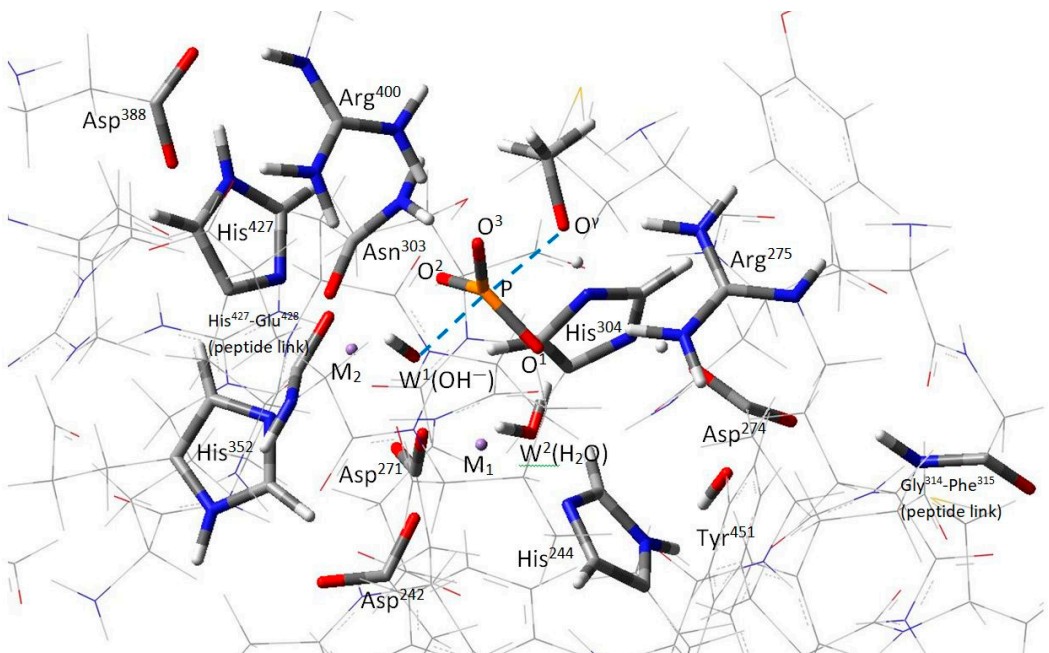

**Figure 2.** Two-level ONIOM(UB3LYP/6-31G(d):UPM7) model of the PP5 catalytic site. The functional groups of the "high-level" region (tubes) are conserved across the PPP gene family; the "low-level" region (wireframe) spans 33 residues. (Only the residues $Gly^{314}$ and $Glu^{428}$ located at the peptide linkages are not conserved.) $M_1$ and $M_2$ are $Mn^{2+}$ ions. The transition state for the backside attack by $W^1(OH^-)$ on methylphosphate dianion (the stand-in substrate) is shown; the substrate's $P$-$O^\gamma$ bond is elongated and the phosphoryl group is flattened as part of a trigonal bipyramidal complex.

In the present work, we used a PP5 co-crystal (pdb entry 4ZX2 [20]) as foundation for a 33-residue model of the PP5 catalytic site (Figure 2). Our goal was to reexamine the $W^1(OH^-)/W^2(H_2O)$ system, and our model was intended to include all relevant moieties that play a role in the reaction and/or serve as steric barriers to prevent nonnative reorientations during optimizations. Two types of calculations were performed: (1) Tests of the behavior of waters associated with the bimetal system and/or protonation states within the catalytic site and (2) tests of the $W^1(OH^-)/W^2(H_2O)$ system for the reactant, transition state, and product stages of the reaction using methylphosphate dianion

($CH_3OPO_3{}^{2-}$) as a stand-in for phosphoserine. The dianionic species was used because kinetic isotope effect data indicated that PPPs act on the dianionic substrate [21] and because the protonated species would have enhanced acidity when metal bound.

## 2. Results

The reactant, TS, and product states for the $W^1(OH^-)/W^2(H_2O)$ system are presented in a truncated form in Figure 3. According to both gas-phase and solvent-corrected energies, the reaction is exothermic ($\Delta H \approx \Delta E = -2.5$ and $-2.0$ kcal/mol), with a low activation barrier ($\Delta H^{\ddagger} \approx \Delta E^{\ddagger} = +8.6$ and $+10.0$ kcal/mol). Vibrational frequency analysis confirms the reactant and product structures as minima and assigns exactly one imaginary frequency ($187i$ cm$^{-1}$) to the TS. Free energy estimates under the rigid rotor/harmonic oscillator approximation afford $\Delta G = -7.2$ and $\Delta G^{\ddagger} = +6.0$ kcal/mol. The coordinates of the optimized structures and an animation of the imaginary mode are available in the Supplementary Materials. $W^2(H_2O)$ helps to stabilize the substrate via H-bonding with $O^1$, and this interaction is present in the TS and product states as well. The other expected stabilizing interactions for the phosphate group are also present in all three states, specifically, $O^2$ with Asn$^{303}$ and $O^3$ with Arg$^{400}$. Importantly, Arg$^{275}$ binds the substrate and TS in a bidentate fashion at $O^1$ and $O^{\gamma}$, just as it binds the phosphate ion in the co-crystal. In short, the reactant state is very similar to the geometry of the PP5/phosphate co-crystal, with no unusual reorientations of the five critical residues that were allowed to move during the optimization. The most significant difference between the co-crystal and the optimized reactant state is that the face of Arg$^{275}$ is turned about 49° to establish an H-bond with Tyr$^{451}$. We add that the reactant state's coordination distance for $W^2$, $r(W^2 - M_1) = 2.323$ Å, is consistent with the X-ray crystal value of 2.295 Å and indicates that the assignment of $W^2$ as a water molecule is correct.

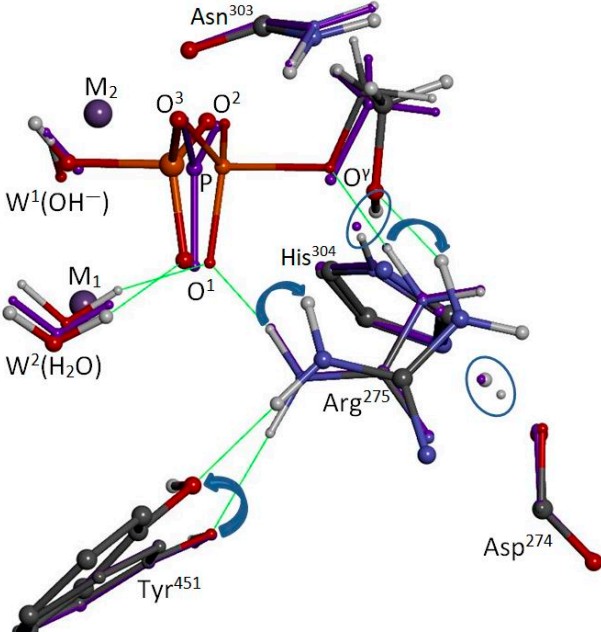

**Figure 3.** Superposition of reactant, transition, and product states of methylphosphate hydrolysis. Representations are small ball-and-stick, violet ball-and-stick, and large ball-and-stick, respectively. Most atoms of the model are hidden from view. $W^1(OH^-)$ at upper left attacks the $M_1/M_2$-bound methylphosphate dianion ($CH_3OPO_3{}^{2-}$), causing rupture of the P-$O^{\gamma}$ bond. Reaction motion primarily involves inversion of the P center and transfer of $H^+$ from His$^{304}$ to $O^{\gamma}$. Post-TS withdrawal of the product alcohol ($CH_3OH$) to the right induces Arg$^{275}$ to swing away from $O^1$, now an atom belonging to the hydrolysis product ($HPO_4{}^{2-}$). At the same time, Tyr$^{451}$ rotates inward to maintain an H-bond with Arg$^{275}$. $W^2(H_2O)$ is H-bonded to $O^1$ throughout the reaction. Proton movements to and from His$^{304}$ are circled; selected H-bonds are indicated by green lines.

As in the previous report [18], the TS is concerted; that is, $W^1(OH^-)$ attacks the phosphorus center at the same time that the exiting seryl/threonyl alkoxide is protonated directly by the $His^{304}$/$Asp^{274}$ tandem. The primary atomic motion over the course of the reaction is the inversion of the P center along with the transfer of $H^+$ from $His^{304}$ to $O^\gamma$. As stated above, $Arg^{275}$ stabilizes the TS in a bidentate fashion through interactions with $O^1$ and $O^\gamma$. Post-TS, however, as the product alcohol recedes, $Arg^{275}$ turns to maintain the $O^\gamma$ interaction, and the $O^1$ interaction is ended. Also post-TS, $Tyr^{451}$ rotates inward to maintain its H-bond with $Arg^{275}$. In the product state, $Arg^{275}$ is isolated from the $HPO_4^{2-}$ product, which is ligated to $M_1$/$M_2$ through $O^1$, $O^2$, and now the bridge hydroxyl oxygen, formerly of the bridge hydroxide $W^1(OH^-)$. As expected with the loss of formal negative charge, the bridge oxygen's metal coordinations are longer in the product state than in the reactant state; meanwhile, the product alcohol lingers, engaged in an H-bonding trio along with $Arg^{275}$ and $His^{304}$.

## 3. Discussion

Crystal structure superposition shows that, in the modeling of Ribeiro et al. [18], $Arg^{275}$ took a position that is partially occupied by $Tyr^{451}$ in the enzyme. This nonnative reorientation occurred because of the absence of $Tyr^{451}$ in their model, and it permitted the transfer of $H^+$ from $Arg^{275}$ to $W^2(OH^-)$. In the enzyme, $Tyr^{451}$ blocks $Arg^{275}$ from contact with $W^2$ when a phosphate-bearing substrate is also ligated to the $M_1$/$M_2$ system. Also, the aforementioned reorientation of $Asp^{274}$ did not occur in our model because we included $Asn^{310}$ and the -NH peptide moiety needed to anchor $Asp^{274}$ (see Figure 1); we froze $Arg^{400}$, but the presence of $Asp^{388}$ in our model (see Figure 2) would have prevented an unrealistic reorientation for $Arg^{400}$ as well. Future improvements to our computational model would allow $Arg^{400}$ and the $M_1$/$M_2$ system to move because of their direct contacts with the substrate, TS, and product phosphate ion. Only subtle changes would be expected, and the states, as shown in Figure 3, should remain qualitatively the same, but the reaction energetics might be impacted. Regarding movement of the $M_1$/$M_2$ system, an increase of about 0.12 Å in the $M_1 \cdots M_2$ separation distance did occur for pathways (I) and (II), upon going from reactant to the transition state(s) [18].

$Arg^{275}$ is mobile and is known to take different positions depending upon the ligand present in the co-crystal [9]. Additionally, we now see that $Tyr^{451}$ interacts with $Arg^{275}$ during the course of the reaction and that their motions are coupled. $Tyr^{451}$ moves to aid the disengagement of $Arg^{275}$ from the product phosphate, which, of course, must be achieved as part of displacing the product phosphate for site regeneration. After the TS, $Tyr^{451}$ follows $Arg^{275}$ while keeping contact with the face of $His^{244}$ and moves only a small distance. Generally, however, $Tyr^{451}$ is part of the flexible β12-β13 loop and is evidently somewhat mobile. Depending upon the ligand present, $Tyr^{451}$ is sometimes vertically farther away from $His^{244}$ and is not always engaged in H-bonding with $Arg^{275}$ in various crystal structures. Interestingly, $PP2A$:$Arg^{89}$ (counterpart of PP5's $Arg^{275}$) can reorient to engage in an apparent salt-bridge interaction at the interface between the catalytic and regulatory domains of PP2A, at least in the presence of microcystin-LR (pdb entry 3FGA [22]). In this orientation, $PP2A$:$Arg^{89}$ still contacts $PP2A$:$Tyr^{265}$ (counterpart of PP5's $Tyr^{451}$). If a natural positioning of $PP2A$:$Arg^{89}$ is truly represented in this crystal structure, it seems reasonable to assume that PP2A does not function efficiently when in such a state.

Quantum-based modeling of some small phosphopeptides as PP5 substrates may be possible in the model we have developed here. Of particular interest, Oberoi et al. [23] investigated the likely positions of the residues adjacent to the p-site serine of Cdc37 ($pSer^{13}$) by appending a mimic sequence (15 residues) to the catalytic domain of PP5. The resulting crystal structure (pdb entry 5HPE) indicates that the backbone –C=O and –NH groups of $Cdc37$:$Val^{12}$ interact with $Arg^{400}$, while the carboxylate of $Cdc37$:$Asp^{14}$ stacks parallel to the face of $Arg^{400}$. Also, the backbone –C=O of $Cdc37$:$Asp^{14}$ interacts with $Arg^{275}$. A substrate longer than a Val-pSer-Asp tripeptide, however, will probably require a larger model of the catalytic site to incorporate the additional contacting residues.

Our results are an improvement over both pathways presented in the earlier computational report by Ribeiro et al. [18], as it is now practical computationally to enlarge the PP5 model system. In their

report, they endorsed the $W^1(OH^-)/W^2(OH^-)$ system and identified $W^2$ as a hydroxide ion: $W^2(OH^-)$. The mechanism they presented, pathway (I), follows an initial $H^+$ extraction from Arg[275] by $W^2(OH^-)$. The charge-zero Arg[275] binds the substrate and TS at $O^1$ but not $O^\gamma$, and its position overlaps with Tyr[451] (absent in the model) in order to interact with $W^2$. In short, Arg[275]'s mode of interaction with the substrate/TS in their model is not optimal for substrate/TS stabilization and does not match the expected bidentate binding seen in the PP5/phosphate complex. Moreover, we argue that an unprotonated arginine ($pK_b \approx 1.5$) that is H-bonded to an electron-rich phosphate moiety would have a very low $pK_b$ and a strong tendency to gain a proton from solvent. Of course, a protonated arginine would also be more effective in drawing the substrate's electron density and, thus, promote hydroxide attack.

The pathway offered in [18] for the $W^1(OH^-)/W^2(H_2O)$ system is endothermic with a high activation energy (>25 kcal/mol), and the assignment option of $W^2(H_2O)$ was consequently discarded [18]. The mechanism presented for this system in [18], pathway (II), involves two steps, with an initial proton transfer to the alkoxide coming from Arg[275], not from His[304]. Instead, in this study of the $W^1(OH^-)/W^2(H_2O)$ system, we have presented a concerted transition state (TS) in which $W^1(OH^-)$ attacks the phosphate center at the same time that the exiting seryl/threonyl alkoxide is protonated directly by the His[304]/Asp[274] tandem. Arg[275], proximal to $M_1$, stabilizes the substrate and TS in a bidentate fashion, and post-TS movement of the adjacent Tyr[451] decouples Arg[275] from the product phosphate ion. The reaction is exothermic ($\Delta H = -2.0$ kcal/mol), as expected, and occurs in one step with a low activation barrier ($\Delta H^\ddagger = +10.0$ kcal/mol).

## 4. Materials and Methods

First, our model was built from pdb entry 4ZX2 [20] (PP5 catalytic domain, 1.23 Å res.) by removal of the inhibitor (a modified norcantharidin) and by selection of proximal residues of the catalytic site. Second, we applied aldehyde (–CHO) and neutral amine (–NH$_2$) terminations where we had broken the enzyme's peptide backbone. Third, hydrogen atoms were added, including one for protonation of His[304]. After placement of the methylphosphate dianion, the bridge hydroxide ($W^1$), and a water of hydration ($W^2$), the added species and all hydrogens were coarsely optimized using the PM7 semi-empirical method as implemented in Gaussian16 [24]. Finally, we selected the atoms of the "high-level" region of our hybrid computational system, as indicated in Figure 2, and carried out ONIOM(UB3LYP/6-31G(d):UPM7) [25–30] partial optimizations. (Of course, depending upon the nature of each computational test, we inserted other waters and removed the hydroxide, the His[304] proton, or substrate, as appropriate.) The system, as shown in Figure 2, has zero total charge. During optimizations, the following residues were allowed to move: His[304], Asp[274], Arg[275], Tyr[451], and Asn[303], as were any added species. The first two residues are the His/Asp tandem. The second two, Arg[275] and Tyr[451], are known to freely move to accommodate ligand binding and were adjusted to approximate their PP5/phosphate co-crystal positions. The last, Asn[303], coordinates $M_2$ and the substrate's phosphoryl group at $O^2$. All calculations were carried out using the Gaussian16 suite of programs [24].

The $Mn^{2+}/Mn^{2+}$ system is an antiferromagnetic singlet state (5 "up"/5 "down"). The proper "high-level" B3LYP electronic state was attained by first constructing an appropriate spin-unrestricted guess wavefunction, as in earlier work [18,31]. The "low-level" PM7 wavefunctions could not be manipulated in the same manner to get antiferromagnetic states in the current implementation of Gaussian16; instead, we converged to stable spin-unrestricted open-shell singlets (1 "up"/1 "down") generated from closed-shell wavefunctions (STABLE = opt). PM7 convergence was aided by using the quadratic convergence option (SCF = yqc). The transition state search was initiated by first carrying out a partial optimization with the inversion dihedral frozen in a nearly planar conformation (GEOM = addgic). Optimizations of the reactant, TS, and product systems were completed using INT = grid = superfine. Optimizations were judged complete when the RMS force value met the default target, as satisfying the full set of default convergence criteria was not practical for a system of this size. Analytic frequencies were computed to confirm the curvature of the energy surface at the two stable

structures and the transition state. Single-point calculations using the default continuum solvation model in Gaussian16 (SCRF = solvent = water, sas) were used to estimate solvent-corrected energies.

**Supplementary Materials:** The following are available at http://www.mdpi.com/2073-4344/10/6/674/s1, File S1: Superposition_of_States.pdb, Video S1: PP5_TS_Animation.gif.

**Author Contributions:** Concept development, E.A.S. and A.W.; computations and draft preparation, E.A.S.; computational resources, A.W.; draft review and editing, A.W. and R.E.H.; project administration and funding acquisition, R.E.H. All authors have read and agree to the published version of the manuscript.

**Funding:** This research and the APC were funded by Grant NIH CA 60750 (R.E.H.).

**Acknowledgments:** This work was made possible in part by a grant of high performance computing resources and technical support from the Alabama Supercomputer Authority.

**Conflicts of Interest:** The authors declare no conflict of interest. The founding sponsors had no role in the design of the study; in the collection, analyses, or interpretation of data; in the writing of the manuscript, and in the decision to publish the results.

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
