# Peer review of "Quantum-Based Modeling of Dephosphorylation in the Catalytic Site of Serine/Threonine Protein Phosphatase-5 (PPP5C)"

_catalysts, doi:10.3390/catal10060674_

Round 1

Reviewer 1 Report

The manuscript entitled “Quantum-based Modeling of Dephosphorylation in the Catalytic Site of Serine/threonine Protein Phosphatase-5 (PPP5C)” described the catalytic process of protein phosphatase. The paper is overall well written with clear presentations of data. Therefore, I suggest its acception in Catalysts after some minor issues are solved.

  1. The authors should provide more information about the metal ions used in their study, such as charge ans size. Is Mn2+/Mn2+ the only pair evaluated? How are different metal ions affecting the final results?
  2. The simulation was done with methylphosphate dianion as a model substrate. However, the actual dephorsphorylation involves much more complicated protein chains. While it might be out of the focus of the current study, it will be helpful if the authors provide some discussions about real protein substrates with larger size.

Reviewer 2 Report

Authors present a cluster model calculation on the dephosphorylation catalyzed by a PP5 enzyme. They overcome previous calculations based on smaller clusters (ref 18) obtaining a concerted mechanism based on W1(OH-)/W2(H2O) arrange of the bimetal active site.

- Figure 2 on page 3: the image background should be white (in my honest opinion), or they could provide an schematic representation of the QM levels.
- Supplementary material is not available (page 5, line 130)

I have a concern about the cluster methodolody: How many residues around are enough?
Authors introduce 33 residues of the active site, which are many more than the used in previous models (10). This is about 6 - 7 Angstroms appart from the substrate in the active site... but when moving up to 13 - 14 Å the total charge of the model becomes negative, which could affect the mechanism (in which a negatively charge moves from one OH- coordinated to a di-cation towards an Asp residue).
Also they perform single energy point calculations using a continuum solvation model (water). Being the active site quite exposed to solvent, wouldn't be worthy to add explicit water molecules?
Finally, they provide enthalpies, but characterize the transition state by the amount of negative frequencies (one), thus why not reporting free energies instead using the RRHO approximation?
